# Research on Defect Detection Method of Fusion Reactor Vacuum Chamber Based on Photometric Stereo Vision

**DOI:** 10.3390/s24196227

**Published:** 2024-09-26

**Authors:** Guodong Qin, Haoran Zhang, Yong Cheng, Youzhi Xu, Feng Wang, Shijie Liu, Xiaoyan Qin, Ruijuan Zhao, Congju Zuo, Aihong Ji

**Affiliations:** 1Institute of Plasma Physics, Chinese Academy of Sciences, Hefei 230031, China; gdqin@ipp.ac.cn (G.Q.); chengyong@ipp.ac.cn (Y.C.); liushijie@mail.ustc.edu.cn (S.L.); 2College of Mechanical and Electrical Engineering, Nanjing University of Aeronautics & Astronautics, Nanjing 210016, China; zhanghaoran1995@126.com (H.Z.); xuyouzhi@nuaa.edu.cn (Y.X.); 3Army Academy of Artillery and Air Defense, Hefei 230031, China; fengwang_1973@126.com (F.W.); xiaoyanqin_hf@163.com (X.Q.); venus_ruijuan@163.com (R.Z.); 4Science Island Branch of Graduate School, University of Science and Technology of China, Hefei 230026, China; 5State Key Laboratory of Mechanics and Control for Aerospace Structures, Nanjing University of Aeronautics and Astronautics, Nanjing 210016, China

**Keywords:** fusion reactor, defect detection, image enhancement, photometric stereo vision, defective reconstruction

## Abstract

This paper addresses image enhancement and 3D reconstruction techniques for dim scenes inside the vacuum chamber of a nuclear fusion reactor. First, an improved multi-scale Retinex low-light image enhancement algorithm with adaptive weights is designed. It can recover image detail information that is not visible in low-light environments, maintaining image clarity and contrast for easy observation. Second, according to the actual needs of target plate defect detection and 3D reconstruction inside the vacuum chamber, a defect reconstruction algorithm based on photometric stereo vision is proposed. To optimize the position of the light source, a light source illumination profile simulation system is designed in this paper to provide an optimized light array for crack detection inside vacuum chambers without the need for extensive experimental testing. Finally, a robotic platform mounted with a binocular stereo-vision camera is constructed and image enhancement and defect reconstruction experiments are performed separately. The results show that the above method can broaden the gray level of low-illumination images and improve the brightness value and contrast. The maximum depth error is less than 24.0% and the maximum width error is less than 15.3%, which achieves the goal of detecting and reconstructing the defects inside the vacuum chamber.

## 1. Introduction

Non-destructive testing as an important part of the safe operation of nuclear power plants has been a concern of scholars around the world [1,2]. For the latest compact fusion devices, due to the large and complex thermal loads and electromagnetic loads, the first wall of the blanket and the target plate area of the divertor are subjected to surface morphology changes, cracks, and other forms of failure during operation [3,4,5], as shown in Figure 1. The length of the cracks and the depth of the defects that appeared in the first wall exceeded the centimeter scale. There are also no relevant tools and algorithms to accomplish this type of detection. To guarantee the high-parameter plasma discharges and complete the risk assessment, precise localization observations and defect reconstruction of the damaged parts must be performed [6,7]. The vacuum chamber of the fusion reactor has an annular design, and the lack of clear light source illumination equipment inside it leads to a very dim scene. To clearly observe the actual condition of the first wall of the vacuum chamber under low illumination, it is necessary to carry out research on low-light image enhancement techniques in the vacuum chamber [8,9,10]. The Retinex theory is an effective method for image enhancement in vacuum chambers [11]. Wang et al. proposed single-scale Retinex, multi-scale Retinex, and multi-scale Retinex with Color Restoration algorithms based on the characteristics of the human visual system [12,13,14]. All these algorithms obtain the illumination component using Gaussian low-pass filtering and utilize only the information from the reflected component for image enhancement [15]. To avoid overexposure, an image enhancement algorithm that preserves naturalness can be constructed. Combining the logarithmically mapped illuminance component with the reflection component can effectively avoid the loss of image information and overexposure phenomenon [16,17]. In addition, naturalness preservation using a priori knowledge of multilayer luminance statistics and using a new filtering approach to extract the illuminance component can also result in smoother low-frequency components [18]. However, Retinex still shows unnatural images in uneven, low-illumination images. Therefore, constraining the illumination component and reflection component to avoid information loss while maintaining the naturalness of the enhanced image is one of the urgent problems to be solved [19,20].

Low-light image enhancement techniques allow observation and localization of surface defects in the first wall of the vacuum chamber. To further analyze the impact of the defects on the next round of experiments in the fusion reactor, 3D reconstruction of the defects is also necessary to assess their damage [21,22]. The photometric stereo vision has the advantages of simple operation, fast measurement speed, and the ability to obtain detailed features on the surface, which are more widely used [23,24]. The 3D information of an image can be recovered from contours, brightness, motion, light and dark, etc., and the light source is not required [25,26,27]. To realize the practical application of photometric stereo vision, many scholars have carried out research on the aspects of image resolution, light source model, material reflection model, calibration method, and so on [28,29]. For the image resolution, Vazquez et al. constructed virtual point-like signal gradient sources using small triangular faces forming polyhedra to achieve super-resolution in images [30]. Scholars have categorized the reflections of the calibration sphere into specular and diffuse reflections for light source calibration. Specular reflections can be used to estimate geometrical parameters, such as the light direction of the light source. In contrast, diffuse reflections can be used to estimate photometric parameters, such as the luminous intensity of the light source [31,32,33]. Based on the multiple viewpoints to obtain the outline information of the measured object and use the interrelationships, the direction and light intensity of the light source can be estimated [34,35]. In the camera-fixed case, introducing irradiation from light sources in different directions increases the constraints on the light sources, and a unique solution can be obtained [36]. The direction of the light source can also be calibrated by comparing the error between the image obtained from the actual light source and the image obtained from the estimated light source [37]. For the 3D reconstruction of defects, Horn et al. proposed the shadow recovery method, which utilizes the shadow, luminance, and reflection information of the object image, and adds other constraints to solve for the depth information [38]. Based on the Fourier basis function 3D reconstruction method, the depth information can be solved by coming up to integrate into the frequency domain and then projecting to the spatial domain using the Fourier inverse transform [39,40]. The complexity of the vacuum chamber structure and the high specular reflection properties of the metal surface make it difficult to measure defects by photometric stereo vision [41,42,43]. How to inspect the surface inside a vacuum chamber based on a reflection model and accomplish 3D reconstruction of defects such as fine cracks is a rarely studied and challenging problem.

This paper addresses image enhancement and 3D reconstruction techniques for dim scenes inside the vacuum chamber of a nuclear fusion reactor. First, a multi-scale Retinex low-light image enhancement algorithm with adaptive weights is designed for the dim interior environment of the fusion reactor vacuum chamber. It can enhance the contrast of low-light images and recover information about details that are not revealed. Second, to meet the needs of defect detection on the surface of the vacuum chamber, a visual servo-based algorithm for defect reconstruction in the vacuum chamber is designed to accomplish the task of reconstructing defects. This paper is divided into 5 sections. Section 2 introduces the low-light image enhancement algorithm for the interior of the vacuum chamber. Section 3 presents the defect reconstruction algorithm based on photometric stereo vision. Section 4 establishes an experimental platform and conducts experiments on image enhancement and defect reconstruction algorithms, and finally, Section 5 gives the conclusion of this paper.

## 2. Low-Light Image Enhancement Algorithm

### 2.1. Image Preprocessing and Threshold Segmentation

Traditional image processing algorithms mainly include image preprocessing, filter denoising, and multi-operator construction of defective edge information. The preprocessing can reduce the non-target information in the image to be detected and highlight the needed image information. The defect information in the internal image of the vacuum chamber can be highlighted by effective threshold segmentation. The gray level of the image to be processed is denoted by r and the gray level range is [0,(L−1)]. Where r=0 denotes black and r=L−1 denotes white, T(r) is a monotonically increasing function on the interval 0≤r≤(L−1). For each pixel point on the input image that has a value r there is a gray scale value s generated. Since the gray level of an image can be regarded as a random variable on the interval [0,(L−1)], let $p_{r}(r)$ and $p_{s}(s)$ denote the probability density functions of random variables r and s, respectively
(1)$$s=T\left(r\right)=(L-1)\int_{0}^{r}p_{r}(w)dw$$
where w is the integral variable and s is the cumulative distribution function of the random variable r. It is easy to get
(2)$$p_{s}(s)=p_{r}(r)\left|\frac{dr}{ds}\right|$$

From the Leibniz criterion it follows that
(3)$$\frac{ds}{dr}=\frac{dT(r)}{dr}=(L-1)p_{r}(r)$$

Substituting Equation (3) into Equation (2) gives
(4)$$p_{s}\left(s\right)=p_{r}\left(r\right)\left|\frac{1}{\left(L-1\right)p_{r}\left(r\right)}\right|=\frac{1}{\left(L-1\right)},\;0\leq s\leq L-1$$

Since Equation (4) is a uniform probability density function, $p_{s}\left(s\right)$ is always uniform and independent of the form of $p_{r}\left(r\right)$. Define the histogram of an image with a gray level range of [0,(L−1)] as a discrete function
(5)$$h\left(r_{k}\right)=n_{k}$$
where $r_{k}$ is the k-th level gray value and $n_{k}$ is the number of pixels with a gray value of $r_{k}$. The total number of image pixels is usually represented by the product MN. Its ratio to each component normalizes the histogram as follows:(6)$$p_{r}\left(r_{k}\right)=\frac{n_{k}}{MN},\;k=0,1,2,\ldots .L-1$$
where L is the number of possible gray levels in the image, and the graph of $p_{r}\left(r_{k}\right)$ relative to $r_{k}$ is called the histogram. Discretizing Equation (6) gives
(7)$$s_{k}=T\left(r_{k}\right)=\frac{\left(L-1\right)}{MN}\sum_{j=0}^{k}n_{j},\;k=0,1,2,\ldots ,L-1$$

The pixels of gray level $r_{k}$ in the input image can be mapped to the corresponding pixels of gray level $s_{k}$ in the output image by Equation (7). Convolutional smoothing using a Gaussian filter gives
(8)$$G\left(x,y\right)=\frac{1}{2\pi \sigma^{2}}e^{\frac{-(x^{2}+y^{2})}{2\sigma^{2}}}$$
(9)$$F\left(x,y\right)=G\left(x,y\right)f\left(x,y\right)$$
where σ is the standard deviation of the Gaussian function, which is used to adjust the degree of smoothing of the image, $f\left(x,y\right)$ is the gray value of the pixel point of the image to be detected, and $F\left(x,y\right)$ is the smoothed image. Image threshold segmentation using Equation (7) has the advantage of being simple, intuitive, fast, and easy to integrate with other algorithms, and the disadvantages of being dependent on threshold settings, unable to deal with complex backgrounds, sensitive to noise, and lack of automatic adjustment mechanism. The results of crack detection inside the vacuum chamber by this method are shown in Figure 2. It is seen that the method has strict requirements on the detection object, which requires obvious image comparison and cannot carry out an in-depth and detailed study of the three-dimensional morphology of the crack defects.

### 2.2. Improved Multiscale Retinex Algorithm

To enhance the contrast of the low-light image inside the vacuum chamber, and at the same time recover the detailed information of the low-light unrevealed image, this paper designs an improved multi-scale Retinex low-light image enhancement algorithm with adaptive weights. The Retinex was initially proposed by Edwin H. Land based on the theory of luminance and color enhancement of human visual perception [11,12], as shown in Figure 3. The detection image inside the vacuum chamber is represented by the binary function f(x,y), the incident light source is called the incident component $i\left(x,y\right)$, and the reflected light is called the reflected component $r\left(x,y\right)$, and the relationship is
(10)$$f\left(x,y\right)=i\left(x,y\right)r\left(x,y\right)$$

The incident component $i\left(x,y\right)$ depends mainly on the light source, which is a low-frequency component, and the reflected component is an intrinsic property of the image. The colour of the surface of the object in the visual system is determined by the reflective ability of the light wave with color invariance. To reduce the amount of computation the Equation (10) is log-transformed
(11)$$R\left(x,y\right)=log\left(\frac{f\left(x,y\right)}{i\left(x,y\right)}\right)=log\left(f\left(x,y\right)\right)-log\left(i\left(x,y\right)\right)$$
where $R\left(x,y\right)$ is the output value of the traditional Retinex algorithm.

From Figure 3, the core of the algorithm is to estimate the incident component $i\left(x,y\right)$ of the original image and remove the reflection component $r\left(x,y\right)$ which is not related to illumination to enhance the overall contrast of the image. To estimate and remove the incident components efficiently, this paper proposes a multi-scale Retinex algorithm with adaptive weights. Setting the incident component as the low-frequency part of the image, by Gaussian low-pass filtering gives
(12)$$R_{MSRi}\left(x,y\right)=\sum_{n=1}^{N}\omega_{n}\left\{\log\left(f_{i}\left(x,y\right)\right)-\log\left(f_{i}\left(x,y\right)G_{n}\left(x,y\right)\right)\right\}$$
where $f_{i}\left(x,y\right)$ is the i-th component of the image, i∈r, denoting the colour channel, respectively. $\omega_{n}$ and $G_{n}\left(x,y\right)$ are the weights and filters corresponding to the n-th multiscale parameter, and the variance is denoted as $\sigma_{n}$. N denotes the number of scales. According to the engineering experience N=3, scale parameters can be taken as $\sigma_{min}=15,\;\sigma_{mid}=80,\;\sigma_{max}=250$. To avoid amplifying noise, color distortion, and halo phenomenon in some cases, different scales can be adjusted during the improvement of the logarithmic function
(13)$$R_{newi}\left(x,y\right)=\sum_{n=1}^{N}\omega_{n}\left\{\log\left[f_{i}\left(x,y\right)+\mu_{1}\right]-\log\left[f_{i}\left(x,y\right)G_{n}\left(x,y\right)+\mu_{2}\right]\right\}$$
where $\mu_{1}$ and $\mu_{2}$ are the incident component and tuning parameter, respectively. The parameter values of $\mu_{1},\mu_{2}$ are usually obtained from image comparison experiments, and in this paper, they are taken as $\mu_{1}=\mu_{2}=9.5$, respectively. To adapt to the flat and uneven regions inside the fusion reactor, the vacuum chamber image is chunked according to the number of scale parameters. The flatness evaluation function inside the vacuum chamber is defined as
(14)$$\rho_{N}=\tau \mathrm{arccot}\left(\sigma_{N}\right)+\upsilon$$
where $\rho_{N}$ is the flatness evaluation index, τ and υ are the adjustment coefficients, and $\sigma_{N}$ is the standard deviation of each sub-block. For flat regions inside the vacuum chamber, the standard deviation of the sub-blocks is smaller, and the value of the exponential function is larger, while the opposite is true for non-flat regions. The standard deviation $\sigma_{N}^{2}$ of the subblock $P_{n}\left(x,y\right)$ partitioned according to the number of scale parameters is
(15)$$\sigma_{N}^{2}=\frac{1}{S}\sum_{\left(x,y\right)\in P_{n}}{\left[P_{n}\left(x,y\right)-\frac{1}{S}\sum_{\left(x,y\right)\in P_{n}}P_{n}\left(x,y\right)\right]}^{2}$$
where S is the total number of pixels in each sub-block. According to the flatness evaluation function, the scale parameter is adjusted
(16)$$\sigma_{N}=\frac{1}{\lambda }\frac{\sigma_{max}-\sigma_{min}}{\rho_{max}-\rho_{min}}(\rho -\rho_{min})$$
where λ is the scale adjustment factor and $\sigma_{min}$ and $\sigma_{max}$ are the minimum and maximum values of each sub-block, respectively. The optimal scale parameter corresponding to each sub-block can be determined by Equation (16) to enhance the brightness while reducing the loss of detailed information. After determining the scale parameters, the weights need to be adaptively adjusted
(17)$$\omega_{m}=\frac{\sum_{m=1}^{N}{\left(\sigma_{max}-\sigma_{m}\right)}^{2}}{3m{\left(\sigma_{N}-\sigma_{m}\right)}^{2}},\;m=1,2,\ldots ,N$$

By combining Equation (16) and Equation (17), the corrected scale parameters and adaptive weights can be determined. Finally, the output value of the Retinex algorithm for the low-light image inside the fusion reactor is
(18)$$R_{fusion}\left(x,y\right)={⋃}_{n=1}^{N}\left\{\sum_{m=1}^{N}\omega_{m}[\log\left(f_{i}\left(x,y\right)+\mu_{1}\right)-\log\left(f_{i}\left(x,y\right)G_{m}\left(x,y\right)+\mu_{2}\right)]\right\}$$
where $R_{fusion}\left(x,y\right)$ is the output result after illumination image enhancement, N is the number of sub-blocks, $\mu_{1}$ and $\mu_{2}$ are the tuning parameters, and $\omega_{m}$ is the weights of the adaptive scale parameters. The larger the weights, the larger the contribution value of the corresponding sub-blocks, and the better the effect of detail preservation and image contrast.

## 3. Defect Reconstruction Based on Photometric Stereo Vision

### 3.1. Modeling Photometric Stereo Vision

The principle of photometric stereo vision is that a set of images of an object individually illuminated by light sources in different directions is captured through the same point of view. According to the luminance equation, the normal vector of the surface of the object, as well as the gradient matrix in the *x* and *y* directions, is calculated to realize 3D stereo reconstruction. The main research core of photometric stereo vision is a 3D reconstruction based on reflections from materials. The ideal diffuse reflection follows Lambert’s cosine law, and its outgoing light intensity is only related to the pitch angle of the incident light
(19)$$\frac{I}{E}=k_{d}cos\theta_{i}=k_{d}\langle l_{i},n\rangle$$
where $k_{d}$ is the diffuse reflection coefficient and $cos\theta_{i}$ denotes the inner product of the normal vector n and the incident light vector $l_{i}$, $\langle l_{i},n\rangle$ denotes the inner product of incident light and surface normal vector. In practice, the bidirectional reflection distribution function (BRDF) can be used to characterize the surface reflection of complex materials. Define the BRDF proportionality between incident irradiance and outgoing radiance in a specified direction as follows
(20)$$\rho_{bd}\left(\omega_{i},\omega_{o}\right)=\frac{dL_{o}(\omega_{o})}{dE_{i}(\omega_{i})}=\frac{dL_{o}(\omega_{o})}{L_{i}(\omega_{i})cos\theta_{i}d(\omega_{i})}$$
where $\omega_{i}=\left(\theta_{i},\varphi_{i}\right)$ is the direction of the incident light, $\omega_{0}=\left(\theta_{0},\varphi_{0}\right)$ is the outgoing light, θ is the altitude angle, φ is the azimuth angle, $E_{i}$ is the irradiance of incident light on the surface of the object, $L_{o}$ is the radiance of the outgoing light on the surface of the object, and $L_{i}$ is the light radiance. For the interior of the vacuum chamber, the BRDF reflection parameters can be obtained by extracting the reflection data from randomly captured images taken by a high-resolution camera at different angles and fitting the model.

According to the Phong reflection model, the intensity of reflected light at any point on the surface of an object is composed of ambient light, diffuse light, and specularly reflected light. Since the vacuum chamber of the fusion reactor is a dark environment, there is no ambient light. Each target plate in the first wall of the vacuum chamber is fabricated with frosted tungsten tiles. The specular reflection is very small, and the specular reflection is concentrated in a small range of reflection angles. With the exclusion of the specular reflection angle, the diffuse reflection model can still be used for approximate calculation. Therefore, a diffuse reflection model can be used to approximate the intensity of the reflected light for the first wall of the vacuum chamber
(21)$$I=\rho \langle l,n\rangle E$$
where $n={\left[n_{x},n_{y},n_{z}\right]}^{T}$ is the unit normal vector at a point inside the vacuum chamber, ρ denotes the reflection coefficient, $l={\left[l_{x},l_{y},l_{z}\right]}^{T}$ is the direction vector of the light source; E is the illuminance of the incident light and I denotes the gray value of the image. Assuming that the light sources in the camera have equal luminosity, i.e., $\rho E=\rho_{1}E_{1}=\rho_{2}E_{2}=\rho_{3}E_{3}$, and the number of light sources is 3, and the matrix consisting of the light source directions is $L={\left[l_{1},l_{2},l_{3}\right]}^{T}$, then Equation (21) is rewritten as
(22)$$I=\rho EL^{T}n$$

From Equation (22), the normal vector at the interior viewpoint of the vacuum chamber can be obtained as
(23)$$n=\frac{L^{-1}I}{\left|L^{-1}I\right|}$$

Assuming that the depth of a point inside the vacuum chamber is z and the gradient is $\left(p,q\right)$, the gradients in its x and y directions are $p=\frac{\partial z}{\partial x}$ and $q=\frac{\partial z}{\partial y}$, respectively. The relationship between the directional gradient and the surface normal vector is
(24)$$n=\frac{{\left(-p,-q,1\right)}^{T}}{\sqrt{p^{2}+q^{2}+1}}$$

The gradient matrix $\left(P,Q\right)$ of all pixels can be obtained by applying the above conversion to the image of the interior of the vacuum chamber acquired by the camera. Assuming that the number of supplementary light sources is k, the incident illumination E inside the vacuum chamber is the same, i.e., $E_{1}=E_{2}=...E_{k}$, $l_{1},l_{2},...{,l}_{k}\in R^{3}$. Then the relationship between the directional gradient and the surface normal vector is
(25)$$n=\frac{{\left(L^{T}L\right)}^{-1}L^{T}I}{\left\|{\left(L^{T}L\right)}^{-1}L^{T}I\right\|}$$
where $L={\left[l_{1},l_{2},...{,l}_{k}\right]}^{T}$, $I={\left[i_{1},i_{2},...{,i}_{k}\right]}^{T}$. The computation of the normal vector inside the vacuum chamber in the case of multiple light sources by photometric stereo vision can be accomplished by Equation (25). The task of reconstructing the surface defects inside the vacuum chamber can be accomplished.

### 3.2. Light Source Positioning Design

During visual inspection inside vacuum chambers, the position of the light source has a significant impact on photometric stereo vision. Sacrificing luminous flux efficiency is often chosen for the uniformity of illumination. That is, less light is provided to the interior of the vacuum chamber to keep the light intensity variation low. Therefore, this paper designs a simulation system of light source lighting contour, and completes the calculation of light source positioning for uniform and efficient lighting by simulating the positional layout of various light sources. The location of the centre of the illumination region of the light source is described in terms of a Cartesian coordinate system, as shown in Figure 4. The normal polar coordinates relative to the vacuum chamber are defined as $\theta_{l}$ and $\varphi_{l}$. By adjusting the number of input variables, the total light flux and illumination variation can be optimized. Based on the input parameter values, the designed optimal light source system can create a set of illumination configurations on its own to calculate the illumination profile observed by the robot. The luminous intensity of the point light source in each direction is
(26)$$I_{\theta }=I_{0}F(\theta )=I_{0}{\left(cos\theta \right)}^{g}$$
where g is the angular attenuation factor related to the uniformity of the light source and $I_{0}$ is the light intensity along the direction of the main optical axis. If g=0, it is an ideal light source; if g=1, it is a Lambertian light source.

The supplementary light source is a hemispherical LED point light source with a diameter of 5 mm, and its luminous intensity and the light intensity obtained on the plane with a distance of d from the LED are
(27)$$I_{LED}\left(\theta \right)=\sum_{i=1}^{n}c_{1i}cos{\left(\theta -c_{2i}\right)}^{c_{3i}}$$
(28)$$E\left(r,\theta \right)=\frac{I_{0}cos\theta }{d^{2}}$$
where $c_{1i}$, $c_{2i}$, $c_{3i}$ are the parameters to be determined and n is the number of cumulative terms of the cosine function. When the reflection characteristics of the material are approximated as Lambertian characteristics, the dual Lambertian illumination model can be obtained
(29)$$E=K_{\alpha }\frac{z^{m+1}L_{LED}A_{LED}}{{\left[{\left(x_{i}-x_{0}\right)}^{2}+{\left(y_{i}-y_{0}\right)}^{2}+z^{2}\right]}^{\frac{m+3}{2}}}=K\frac{z^{m+1}}{{\left[d^{2}+z^{2}\right]}^{\frac{m+3}{2}}}$$
where $K_{\alpha }$ is the camera sensor response parameter, $L_{LED}$ is the luminous brightness of the LED, $A_{LED}$ is the area of the light emitting chip of the LED, and the value of m under the Lambertian light source is 1. *d* is the distance between the point light source and the measured position of the vacuum chamber, and *z* is the depth value of the measured position. The task of completing the optimal lighting configuration of the system according to the designed efficient light source positioning system is shown in Table 1, and the output is shown in Figure 5. Related filtering optimizations can be accomplished by testing with multiple sets of instances. By applying Lambert’s cosine emission law and the distance factor from the secondary target to the lens, the final optical radiation intensity can be determined. In addition, illumination from multiple light sources can be efficiently calculated using positional symmetry. The optimal light source positioning system designed in this paper can broaden the setup output results to provide optimized light arrays for crack detection inside vacuum chambers without the need for time-consuming experimental testing.

## 4. Defect Detection Experiments

### 4.1. Experimental Platform

The vacuum chamber robot hardware system is shown in Figure 6a, which mainly consists of an 11-degree-of-freedom robot, an end binocular camera, and a light source system. The control system consists of a high-performance multi-axis motion control card and a NVIDIA Jetson Nano control board, which brings sufficient arithmetic power for the end processing of the 3D reconstruction of the fissure. The camera is a RGBD binocular camera, and its specific solid-state parameters are shown in Table 2. The experimental object of the vacuum chamber defect detection designed in this paper is a composite matte plate, whose length, width, and height are divided into 400 mm×400 mm×20 mm, as shown in Figure 6b. To reduce the randomness error of detection, the actual depths of the three defects on the simulated plate in the vacuum chamber selected for the experiment are 15 mm, 10 mm and 5 mm, respectively.

The software system of the fusion robot inspection algorithm mainly consists of a path planning module and a defect detection module, as shown in Figure 7. The 3D path planning program of the fusion robot is then realized by calling the function functions directly in the motion control card. The defect detection system for the vacuum chamber is mainly based on a VMware virtual machine for wireless communication. In the constructed wireless local area network (WLAN), Putty-0.67 software is utilized for the serial interface connection task. The camera carried at the end of the robot can continuously acquire images of the interior of the vacuum chamber during the inspection process. Relevant depth information and 3D construction of defects inside the vacuum chamber can be accomplished in real-time.

### 4.2. Low-Light Image Enhancement Experiment

To verify the effectiveness of the image enhancement algorithm proposed in this paper, the defects inside the vacuum chamber are first detected using traditional methods. The detection steps mainly include smoothing processing, threshold segmentation and, defect contour extraction, etc. The experimental results of contour information detection of simulated defects inside the vacuum chamber are shown in Figure 8. The analysis results show that the contour length information can be obtained by setting the input image average brightness and overall size parameters, and the defect contour extraction is initially realized. The results of depth information detection are shown in Figure 9. The depth of detection takes the colour of the hue, saturation, and value (HSV) space, and the different colours indicate the proximity to the location of the detection camera. From the experimental results, it can be seen that the reconstructed crack location and the actual location are different, and there is a large amount of noise in the detection results at different moments. The actual defect location of the simulated plate is 600 mm away from the end camera of the robot, and the experimental results show that the blue area indicates a depth value of 620 mm and the red area indicates a depth value of 640 mm, while the actual defect depth value is between 630 mm and 650 mm. It can be seen that the traditional method has a large error in the detection results, and only obtains the approximate distance of the detection target, which is unable to realize the accurate defect reconstruction task. And when there is a low-light situation, the method is almost completely ineffective.

To this end, an improved multi-scale Retinex algorithm with adaptive weights is designed in this paper for low illumination image enhancement, and the experimental results are shown in Figure 10. Figure 10a–d shows the original illuminated images of the experimental scene and the simulated defective plates. Figure 10e–h shows the results of the low-illuminated images after enhancement. It can be observed that the multi-scale Retinex algorithm with improved adaptive weights designed in this paper can enhance the contrast of the image and maintain the detailed information with no additional added noise. From Figure 10h, it can be observed that all three defect information can be detected using the algorithm designed in this paper.

To facilitate the observation of the enhancement effect, the images before and after enhancement are processed with normalized image gray-level histogram, as shown in Figure 11. Where Figure 11a–d shows the normalized gray level histogram of the images before enhancement, it can be observed that the overall gray level of each image is low and concentrated in the low gray level region. Figure 11e–h shows the normalized grayscale histogram of the image after the enhancement process, and it can be seen that the grayscale levels have all been broadened to varying degrees, the overall luminance values have been enhanced, and the contrast has also been improved, which proves the effectiveness of the image enhancement algorithm.

### 4.3. Defect Reconstruction Experiment

When the low-light image is too blurred, even with the enhancement algorithm, it is still not possible to obtain more defect information, especially the detection of small defect information. To accurately reconstruct the defect information, this paper uses three parallel light sources in different directions to increase the light illumination of the vacuum chamber and carries out 3D reconstruction of defects based on the principle of photometric stereo vision. The incident unit vectors of the three light sources are $l_{1}$, $l_{2}$, and $l_{3}$. By lighting up the light sources one by one, the gray values $I_{1}$, $I_{2}$, and $I_{3}$ of the three images under different illumination can be obtained, and the value of the normal vector n can be obtained through the calculation in Section 3.1. The experimental detection process with three light sources acting individually is shown in Figure 12.

Since the vector product of the normal to the simulated plate and the light source is proportional to the irradiance, from this, we can obtain the ratio of the intensity of the light sources of #1, #2, and #2 #3 to each other. In this paper, the scatter interpolant function is utilized to store the above two ratios and to build a light intensity lookup table, where the illuminance under the action of light source #1 is 215 lux, the illuminance under the action of light source #2 is 3380 lux, and the illuminance under the action of light source #3 is 577 lux. The value of the gradient at a point on the detection image has been defined as (p,q) in Section 3.1 and used as a function of the image intensity. Set q and p to range from −10 to 10 in steps of 0.1. Since a lookup table has been constructed to find the gradient (p,q) for each pixel, a gradient map can be constructed with the same size of the defective image.

The distance between the camera and the specimen in the actual detection experiment is 400 mm, in which the measured values of the width and the depth are taken as the maximum of the contour extraction and the 3D reconstruction. The gradient and normal direction of each pixel inside the vacuum chamber can be efficiently calculated using the lookup table. Then, integration is performed along a path to recover the relative distance of each pixel of the defect to complete the 3D reconstruction of the cracked defect in the image, and the results are shown in Figure 13 and Table 3. Where x is the length of the defect in the horizontal direction, y is the length of the defect in the vertical direction, and z is the depth of the vertically detected plate. Detailed location and depth information of defects can be obtained from the experimental results. The maximum depth error is less than 24.0%, and the maximum width error is less than 15.3%. There is a certain degree of aberration in the shape of defects, mainly due to the following reasons: (1) the condition of integrable edges in photometric stereo vision is restricted; (2) the camera’s viewing angle limitation, which makes the defect location produce more shadows and highlights areas leading to errors in the gradient map establishment. In general, the gradient maps of the vacuum chamber defect detection are well established, and the goal of defect recognition and reconstruction is accomplished.

## 5. Conclusions

In this paper, an improved multi-scale Retinex low-light image enhancement algorithm with adaptive weights is designed to detect low-light images inside the fusion reactor vacuum chamber. It can not only enhance the image contrast in the low illumination situation inside the vacuum chamber but also recover the detailed information of the image that is not shown in the low illumination situation, and effectively maintain the clarity and contrast of the image. A defect detection system based on photometric stereo vision is constructed for defect detection and 3D reconstruction inside vacuum chambers. The normal vector and gradient field of the object surface are calculated from the illumination direction and the light/dark relationship of the vacuum chamber image. The gradient matrix of all the pixels on the image is obtained by matrix transformation, which ultimately accomplishes the task of reconstructing the 3D depth matrix. To optimize the position of the light source, a simulation system is designed for a given light source illumination profile. The optimized light array can be provided for crack detection inside vacuum chambers without time-consuming experimental testing. Finally, a vacuum chamber inspection robot platform mounted with a binocular stereo vision camera is built, and image enhancement and defect reconstruction experiments are performed separately. The results show that the multi-scale Retinex low-light image algorithm based on adaptive weights can broaden the gray level of low-light images and increase the brightness value and contrast. The defect reconstruction algorithm based on photometric stereo vision can obtain the detailed position and depth information of different defects, with a maximum depth error of less than 24.0% and a maximum width error of less than 15.3%, which realizes the task objectives of detecting and reconstructing surface defects inside the vacuum chamber.

In the future, we will continue researching defect detection algorithms for fusion reactor vacuum chambers. The defect detection experiments under different illumination will be conducted through the China Fusion Engineering Test Reactor, which is currently under construction, to verify the effectiveness of the image enhancement and defect reconstruction algorithms.

## Figures and Tables

**Figure 1 sensors-24-06227-f001:**
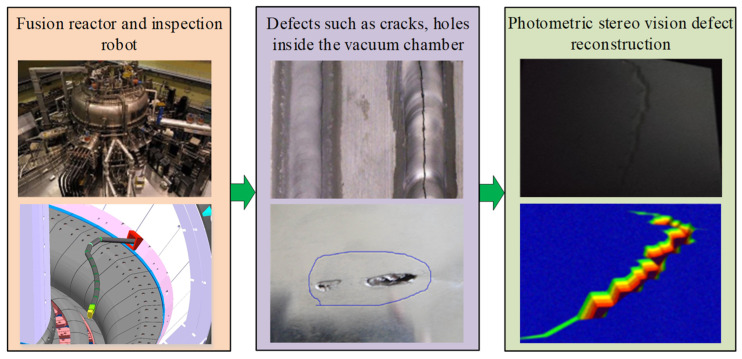
Fusion reactor vacuum chamber defect detection.

**Figure 2 sensors-24-06227-f002:**
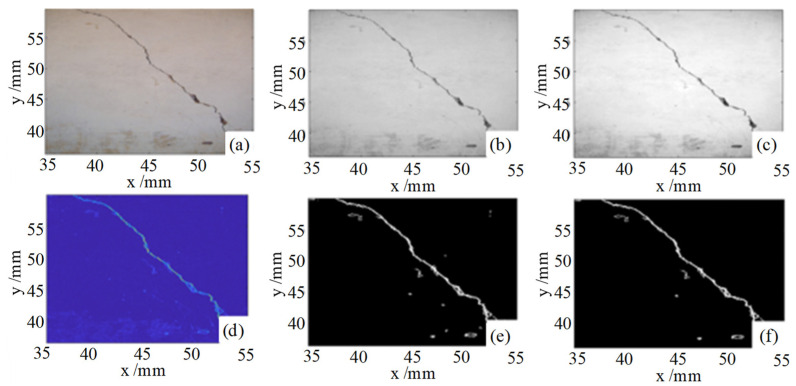
Vacuum chamber defect detection (**a**) original image of cracks (**b**) grayscale image (**c**) Gaussian filtered image (**d**) standard gradient map (**e**) binary map (**f**) image with noisy regions removed.

**Figure 3 sensors-24-06227-f003:**
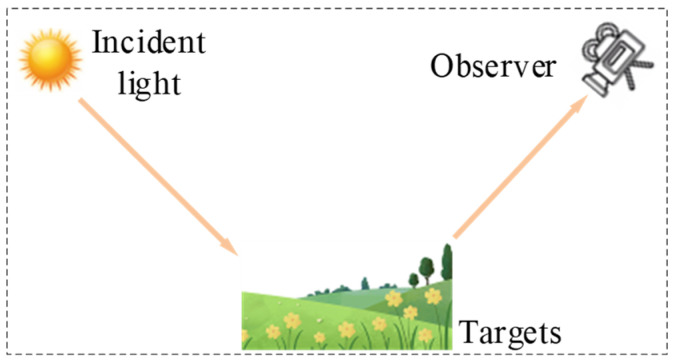
The principles of the human visual system.

**Figure 4 sensors-24-06227-f004:**
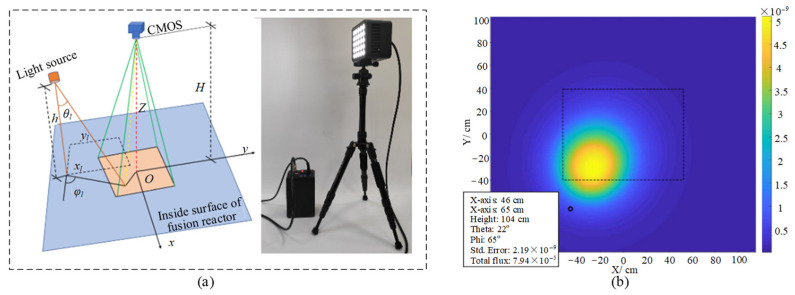
Light source location and illumination profile (**a**) LED light source relative position and variables (**b**) single light source contour.

**Figure 5 sensors-24-06227-f005:**
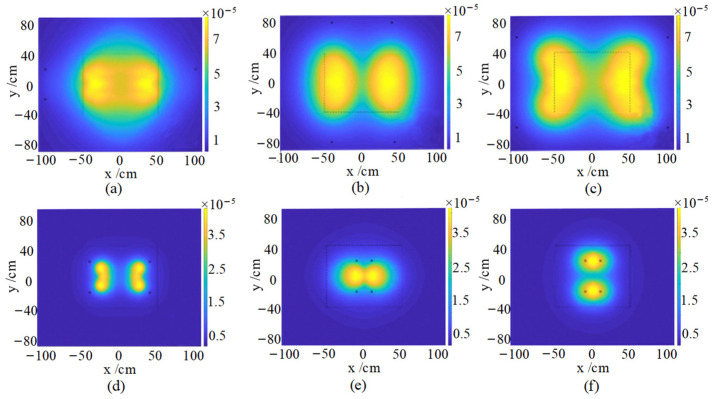
Light source location and illumination profile for multi-input tests (**a**) light 1: $\theta_{1}=0{}^{\circ},\;\alpha_{1}=0{}^{\circ}$ (**b**) light 2: $\theta_{1}=30{}^{\circ},\;\alpha_{1}=30{}^{\circ}$ (**c**) light 3: $\theta_{1}=30{}^{\circ},\;\alpha_{1}=30{}^{\circ}$ (**d**) light 4: $\theta_{1}=30{}^{\circ},\;\alpha_{1}=90{}^{\circ}$ (**e**) light 5: $\theta_{1}=60{}^{\circ},\;\alpha_{1}=30{}^{\circ}$ (**f**) light 6: $\theta_{1}=30{}^{\circ},\;\alpha_{1}=30{}^{\circ}$.

**Figure 6 sensors-24-06227-f006:**
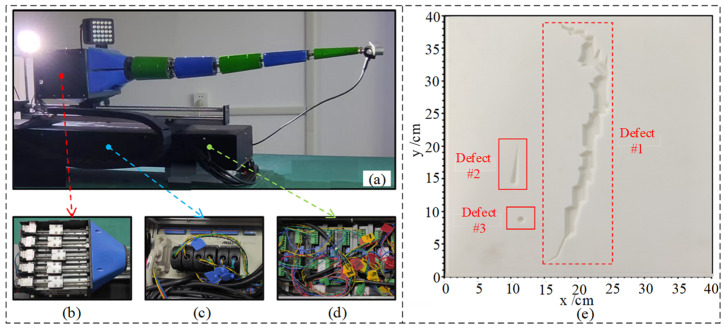
Experimental platform for detecting defects (**a**) robot (**b**) drive Box (**c**) wiring terminals (**d**) drivers (**e**) defects of setting.

**Figure 8 sensors-24-06227-f008:**
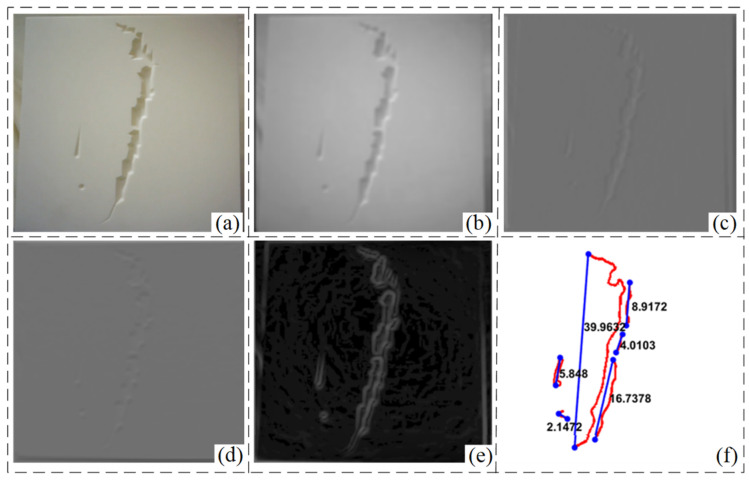
Detection results of internal defects by conventional method (**a**) original image (**b**) smoothing (**c**) horizontal detection (**d**) vertical detection (**e**) defect extraction (**f**) contour measurement.

**Figure 9 sensors-24-06227-f009:**
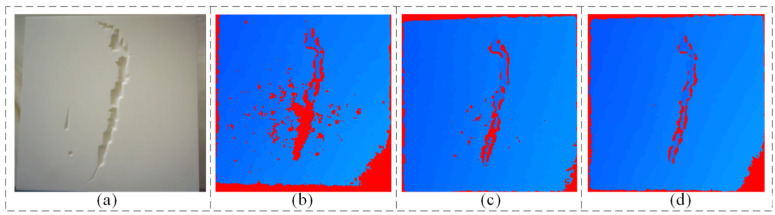
Defect depth maps at different moments by conventional method (**a**) defective samples (**b**) depth map at t = 0 s (**c**) depth map at t = 10 s (**d**) depth map at t = 30 s.

**Figure 10 sensors-24-06227-f010:**
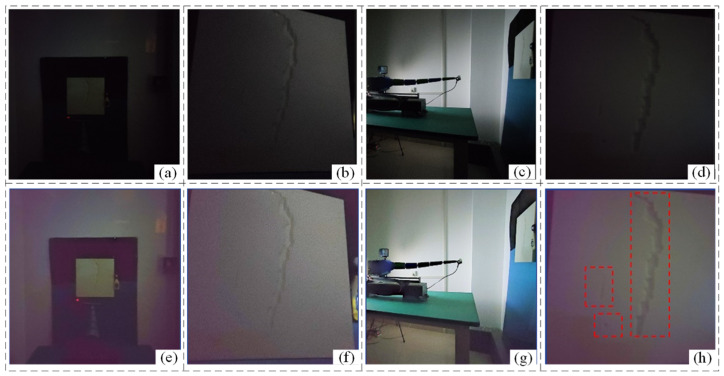
Experiments on low-light image enhancement of cabin surface by improved method (**a**) traditional method main view (**b**) traditional method low illumination (**c**) traditional method side view (**d**) traditional method very low illumination (**e**) improved method main view (**f**) improved method low illumination (**g**) improved method side view (**h**) improved method very low illumination.

**Figure 11 sensors-24-06227-f011:**
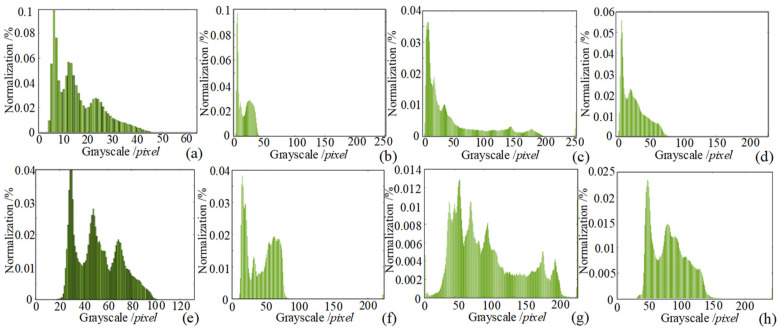
Normalized histograms of image enhancement experiments (**a**) main view histogram (**b**) low illumination histogram (**c**) side view histogram (**d**) very low illumination histogram (**e**) main view of improved method (**f**) low illumination of improved method (**g**) side view of improved method (**h**) very low illumination of improved method.

**Figure 12 sensors-24-06227-f012:**
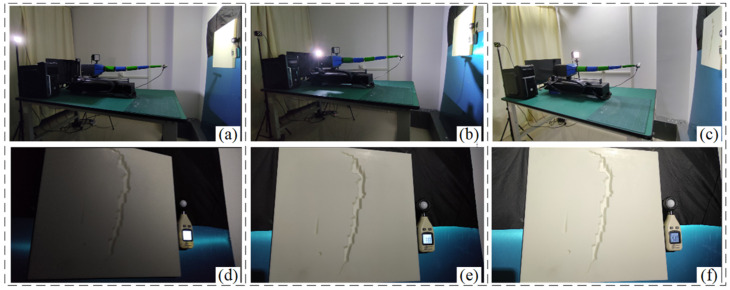
Detection results under different LED light sources (**a**) #1 light source detection experiments (**b**) #2 light source detection experiments (**c**) #3 light source detection experiments (**d**) #1 light source detection image (**e**) #2 light source detection image (**f**) #3 light source detection image.

**Figure 13 sensors-24-06227-f013:**
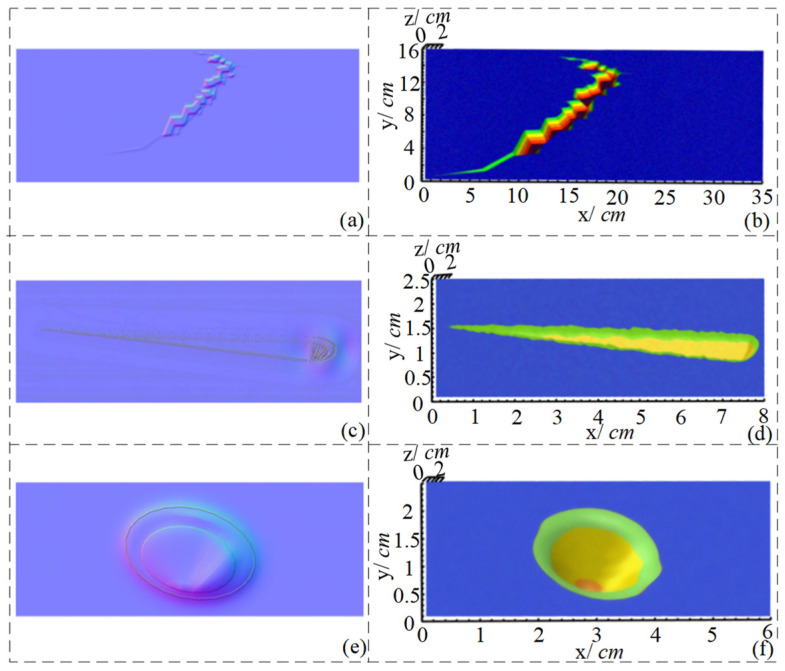
Surface defect normal vectors and 3D reconstruction results (**a**) Normal vector map of defect #1 (**b**) 3D reconstruction results of defect #1 (**c**) Normal vector map of defect #2 (**d**) 3D reconstruction results of defect #2 (**e**) Normal vector map of defect #3 (**f**) 3D reconstruction results of defect #3.

**Figure 7 sensors-24-06227-f007:**
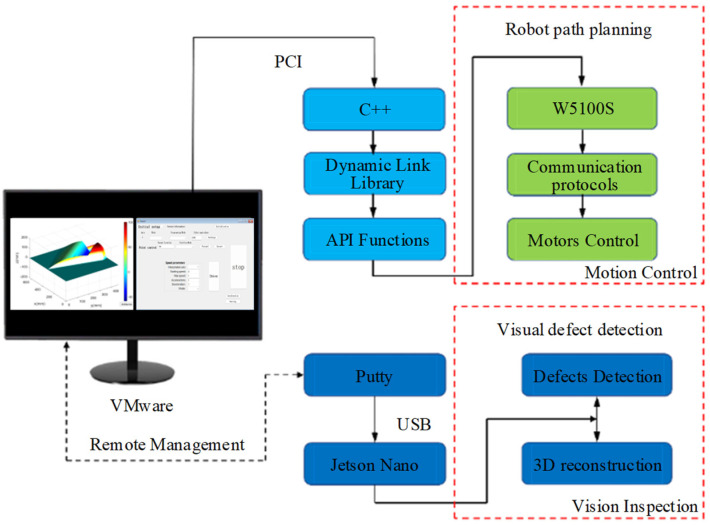
Control system for defect inspection platform.

**Table 1 sensors-24-06227-t001:** Input values for the light source positioning system.

LED Light	*X*-Axis	*Y*-Axis	Height	$\theta_{l}$	$\varphi_{l}$
Light 1	10 cm	20 cm	60 cm	0°	0°
Light 2	10 cm	20 cm	60 cm	30°	30°
Light 3	40 cm	20 cm	35 cm	30°	30°
Light 4	40 cm	80 cm	110 cm	30°	90°
Light 5	100 cm	20 cm	60 cm	60°	30°
Light 6	100 cm	60 cm	110 cm	30°	30°

**Table 2 sensors-24-06227-t002:** RGBD binocular camera solid state parameters.

Types	Parameters
Camera Size	59 mm×17 mm×11 mm
Depth accuracy Depth resolution	1 m ± 6 mm 640 × 400
Deep field of view	H: 67.9° V: 45.3°
RGB field of view	H: 71.0° V: 43.7°
Baseline	120.0 mm
Monitoring range	2.1 mm
Synchronization accuracy	<1 ms
Scope of work	0.8 m–5 m

**Table 3 sensors-24-06227-t003:** Defect detection results of specimens inside the vacuum chamber.

No.	Depth	Width	Depth Error	Width Error
#1	11.57 mm	34.6 mm	22.9%	15.3%
#2	8.3 mm	11.9 mm	17.0%	12.5%
#3	3.8 mm	16.3 mm	24.0%	8.7%

## Data Availability

The data that support the findings of this study are available from the corresponding author upon reasonable request.

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
