# Peer review of "Research on Defect Detection Method of Fusion Reactor Vacuum Chamber Based on Photometric Stereo Vision"

_sensors, 2024, doi:10.3390/s24196227_

Round 1
Reviewer 1 Report
Comments and Suggestions for Authors
This manuscript introduces an improved multi-scale Retinex algorithm for low-light image enhancement, combined with photometric stereo vision technology, for defect detection and 3D reconstruction inside the vacuum chamber of a fusion reactor. The article has made certain advancements in image enhancement and defect reconstruction, especially in low-light environments. In summary, major revision is still required before full acceptance.
(1) The article proposes a detection method that combines a multi-scale Retinex algorithm with photometric stereo vision, targeting the problem of defect detection inside the vacuum chamber of a fusion reactor. This topic is closely aligned with the current technical needs of the industrial inspection field and has high practical application value. However, the article slightly lacks innovation, as the technical solution proposed has similar studies in existing literature. It is recommended that the authors further highlight the differentiated features of this study from existing technologies, clarify the innovative points of this research, and specify the concrete advantages over existing technologies.
(2) The overall structure of the article is relatively complete, with a reasonable arrangement of content from problem background, method proposal, experimental validation to conclusion summary. However, the logic in some sections needs to be strengthened. Especially in the method introduction part, the derivation process of the algorithm is not clearly described, which may cause difficulties for readers to understand the algorithm.
(3) The proportion of references from the last five years is not high, and some of the cited references are not strongly related to the article's topic. It is recommended that the authors pay more attention to the specificity and relevance when citing references, and choose literature that is directly related to the research topic to enhance the persuasiveness of the article.
(4) In the introduction, the first paragraph should present a broader and more comprehensive view of the problems related to the research topic, with citations to authority references (SUPPOSe 3Dge: A Method for Super-Resolved Detection of Surfaces in Volumetric Fluorescence Microscopy; Journal of Optics and Photonics Research. 3D vision technologies for a self-developed structural external crack damage recognition robot; Automation in Construction.).
(5) In terms of technical depth, the details of some key technologies are not fully described, such as the specific implementation process of light source positioning design and defect reconstruction algorithms. In terms of technical breadth, there is insufficient discussion on the applicability and limitations of the algorithm. It is suggested that the authors further deepen the discussion of technical issues, provide more technical details, and comprehensively analyze the advantages and disadvantages of the algorithm.
(6) In the introduction, the second paragraph feels like a mere listing, lacking logical coherence and progressive relationship in the narration.
(7) For computer vision applications, please refer to A lightweight improved YOLOv5s model and its deployment for detecting pitaya fruits in daytime and nighttime light-supplement environments; Computers and Electronics in Agriculture. Dual-Frequency Lidar for Compressed Sensing 3D Imaging Based on All-Phase Fast Fourier Transform; Journal of Optics and Photonics Research.
(8) In the low-light image enhancement algorithm chapter, there appear to be extra spaces in formulas (12), (13), (14), and (15), and "log," "arccot," and "cos" should not be in italics. Formula (15) has a formatting issue that requires careful checking. In the process of describing image preprocessing and threshold segmentation, the derivation of formula (7) seems overly simplified and does not adequately consider the issues of image noise and uneven illumination. The authors need to further explain the effectiveness and limitations of this formula in practical applications.
(9) In the algorithm description, the weight adjustment strategy in formula (13) appears to be disconnected from the actual image content. The calculation of weights is based on the local variance of the image but does not take into account the specific content and complexity of the scene. For instance, in areas of high reflection or low texture, this approach could lead to inappropriate weight distribution, thereby affecting the final image enhancement effect. Although an adaptive weight multi-scale Retinex algorithm is proposed, the paper does not elaborate on the specific implementation details of the weight adjustment strategy.
(10) In the chapter on defect reconstruction based on photometric stereo vision, the authors mention the use of BRDF to describe the surface reflection of complex materials, but do not provide an estimation method for the BRDF parameters. The article designs a light source illumination contour simulation system, but in the experimental design, it does not fully consider the impact of the light source position on the detection results. It is recommended that the authors experimentally verify the specific impact of different light source configurations on the performance of defect detection.
(11) In the defect detection experiments chapter, the clarity of Figure 7 needs to be enhanced. Although the experimental section demonstrates the image enhancement effect of the algorithm under low-light conditions, it lacks comparative experiments with existing technologies. The authors should provide comparative experiments with one or more existing algorithms to prove the superiority of the proposed method. In Figure 13, there are significant differences between the reconstructed results of some defects and their actual shapes. This may be due to the insufficient accuracy of the algorithm when dealing with complex surfaces or small-sized defects. Furthermore, the error analysis of the reconstruction results is not detailed enough and fails to fully explain the causes of the errors.
Author Response
Thanks for the reviewer's comments; we have responded to each question and carefully revised the manuscript, which is highlighted in red. The file of responses to reviewers has been uploaded to the attachment, please check it. And if you have any more questions, feel free to give us feedback, and we'll revise it again.

Reviewer 2 Report
Comments and Suggestions for Authors
The authors propose a defect detection method for fusion reactor vacuum chambers using a photometric stereo approach. To enhance image quality in low-light conditions, the authors employ an improved multi-scale Retinex algorithm with adaptive weights. However, some major issues must be addressed before acceptance.
1. The paper utilizes an improved Retinex algorithm for low-light image enhancement. However, this method may alter the shading information critical for photometric stereo reconstruction, which heavily relies on accurate intensity values. Given the potential non-linear changes introduced by the enhancement algorithm, especially on materials with varying reflectance properties, the authors have to conduct ablation studies to verify the impact of this enhancement on the reconstruction accuracy.
2. The authors do not seem to have a clear understanding of the principles of photometric stereo. Simply put, PS derives surface normals by utilizing shading variations across images taken under different lighting conditions, rather than the intensity values themselves. Given this, what is the purpose of performing low-light enhancement?
3. The authors have employed a fundamental photometric stereo technique. However, in industrial inspection, the targets are often metallic materials with complex reflective properties, including non-Lambertian surfaces (not sure of the material of the fusion reactor vacuum chamber, sorry if it is a pure Lambertian material). How does the proposed method ensure accuracy under such conditions? Have the authors considered the impact of surface reflectance characteristics on reconstruction accuracy, and are there any specific strategies or model improvements to address these challenges?
4. Did the authors use parallel light sources in their experiments? Were the experiments conducted in a standard darkroom environment? If so, how can such stringent experimental conditions be applied to real-world industrial inspection scenarios? The authors should consider recent work such as Universal PS [1], which explores the adaptability and robustness of photometric stereo methods under non-ideal conditions.
5. What method did the authors use to integrate the depth information obtained from binocular stereo vision with the normal information derived from the photometric stereo method? It seems there are no details in the manuscript, sorry if I miss them. What is the specific purpose of using binocular stereo vision for depth acquisition, given that the experimental results seem to rely primarily on normal maps? Could the authors clarify the role and contribution of binocular depth information in the overall methodology?
6. The references are quite old, actually, there are lots of recent methods in Photometric Stereo and others, such as the report in [2] and [3]. I suggest the authors should fully do the related work section.
[1] Scalable, Detailed and Mask-free Universal Photometric Stereo, CVPR 2023
[2] Deep Learning Methods for Calibrated Photometric Stereo and Beyond, TPAMI 2024
[3] A benchmark dataset and evaluation for non-lambertian and uncalibrated photometric stereo, TPAMI 2019
Author Response
Thanks for the reviewer's comments; we have responded to each question and carefully revised the manuscript, which is highlighted in red. The file of responses to reviewers has been uploaded to the attachment; please check it. And if you have any more questions, feel free to give us feedback, and we'll revise it again.

Round 2
Reviewer 1 Report
Comments and Suggestions for Authors
accept
Author Response
Thank you for your comment.
Reviewer 2 Report
Comments and Suggestions for Authors
Thank you for your responses. I am sorry for my misunderstanding. Although the current version of the manuscript meets the publication standards of Sensors, I still believe the following two issues are critical:
While I understand your clarification regarding the two-step process of applying the low-light enhancement algorithm followed by defect detection using photometric stereo (PS), I believe this could be better explained in your abstract and introduction. Please ensure that it is clearly stated that the low-light enhancement occurs first, followed by PS for defect recognition.
Although you mentioned using the BRDF model to describe non-Lambertian surfaces, your current explanation is insufficient. Could you specifically indicate whether you are using the Phong reflection model or another model? Additionally, from Eqs. 23-25, it seems that you are still using the least squares method, which does not fully account for non-Lambertian reflectance properties. Actually, in Eq. 22, the authors reckon the \rho as a scalar therefore the author got Eq. 23, which is not true in real non-Lambertian surfaces. This contradicts the author's response Q3.
Author Response

(The authors gave the same response as above.)
